# Association between Coronary Artery Plaque Progression and Liver Fibrosis Biomarkers in Population with Low Calcium Scores

**DOI:** 10.3390/nu14153163

**Published:** 2022-07-30

**Authors:** Tsung-Ying Tsai, Pai-Feng Hsu, Cheng-Hsueh Wu, Shao-Sung Huang, Wan-Leong Chan, Shing-Jong Lin, Jaw-Wen Chen, Tse-Min Lu, Hsin-Bang Leu

**Affiliations:** 1Division of Cardiology, Department of Cardiovascular Medicine, Taichung Veterans General Hospital, Taichung 407, Taiwan; johntasi222@gmail.com; 2Division of Cardiology, Department of Medicine, Taipei Veterans General Hospital, Taipei 112, Taiwan; chwu32@gmail.com (C.-H.W.); sshuang2@vghtpe.gov.tw (S.-S.H.); phys@vghtpe.gov.tw (W.-L.C.); sjlin@vghtpe.gov.tw (S.-J.L.); jwchen@vghtpe.gov.tw (J.-W.C.); tmlu@kimo.com (T.-M.L.); 3Healthcare and Management Center, Taipei Veterans General Hospital, Taipei 112, Taiwan; 4Faculty of Medicine, National Yang-Ming Chiao Tung University, Taipei 112, Taiwan; 5Institute of Clinical Medicine and Cardiovascular Research Center, National Yang-Ming Chiao Tung University, Taipei 112, Taiwan

**Keywords:** nonalcoholic fatty liver disease, coronary artery disease, coronary computed tomography angiography, metabolic biomarkers

## Abstract

Background: The severity of nonalcoholic fatty liver disease (NAFLD) has been found to be associated with atherosclerosis burden. However, whether liver fibrosis scores can be used to predict atherosclerosis progression, especially for patients with low calcium scores, remains undetermined. Methods: A total of 165 subjects who underwent repeated coronary computed tomography angiography (CCTA) and had low calcium scores (<100) were enrolled. The segment stenosis score (SSS) from the CCTA was measured, and the association between SSS progression and biochemical parameters was analyzed in addition to liver fibrosis scores, including nonalcoholic fatty liver disease fibrosis score (NFS), fibrosis-4 index (FIB-4), aspartate aminotransferase (AST) to platelet ratio index (APRI), and Forns score. Results: When compared with those without plaque at baseline (SSS = 0), subjects with plaque had higher blood pressure, higher coronary artery calcium (CAC) scores, and higher liver fibrosis scores, including Forns score, Fib-4, and NFS. During the medium follow-up interval of 24.7 months, 60 (39.4%) patients displayed SSS progression, while the remaining 105 (63.6%) patients showed no CAD progression. In a multivariate analysis, being male having a high diastolic blood pressure (DBP), and having a high NFS liver fibrosis score were independently associated with the odds ratio for SSS progression. Conclusions: Higher baseline blood pressure and liver fibrosis markers are associated with the presence of coronary artery disease (CAD) plaques in subjects in early CAD stages. For disease progression, the male gender, DBP, and NFS appear to be independently associated with coronary atherosclerosis plaque progression in subjects with low calcium scores.

## 1. Introduction

Coronary artery disease (CAD) is a major health risk affecting millions of people worldwide [1]. Coronary computed tomography angiography (CCTA) is the most powerful noninvasive tool for the detection of the coronary plaque burden and plaque characteristic evaluations [2]. The coronary artery calcium score (CAC) is the backbone of CCTA risk stratification and has been used as a prognostic tool for CAD [3]. CAC does not require a contrast medium injection and can be estimated from noncardiac CT protocols, making it an ideal tool for screening and risk stratification for CAD. According to the current guidelines, patients with low or zero CAC values are said to have a lower risk of future cardiovascular (CV) events [4]. However, a low CAC is not risk-proof, as many patients with a low CAC still suffer from CV events [5,6]. It has been observed that 7%–38% of obstructive CAD is found in patients with a CAC score of 0, especially among those with angina symptoms [7]. Furthermore, CAD is a dynamic disease, and in the multiethnic study of atherosclerosis (MESA), more than 50% of patients with zero CAC had progressive coronary atherosclerosis after the first decade [8]. The progression of atherosclerosis as shown by CCTA is associated with future CV events that are independent of traditional risk factors [9,10]. Although traditional CAD risk factors, such as age, gender, ethnicity, hypertension, type 2 diabetes mellitus (DM), hyperlipidemia, and overweight/obesity, are associated with plaque progression according to CCTA [11], these risk factors can only explain a fraction of the CAD risks. Many nontraditional cardiometabolic biomarkers have been investigated in the field of CAD [12]. Among them, nonalcoholic liver fibrosis markers, a noninvasive tool initially developed to assist with the diagnosis of nonalcoholic liver disease, have also been found to be associated with CAD [13]. Nonalcoholic fatty liver disease (NAFLD) is the most common chronic liver disease and a leading cause of cirrhosis globally [14]. In addition to being a liver disease, NAFLD is an important risk factor for DM, metabolic syndrome, and cardiovascular (CV) diseases [15]. NAFLD shares many pathophysiological mechanisms with CAD, and the progression of NAFLD as measured by the progression of liver fibrosis scores has been associated with both a worse CAD burden and worse CV outcomes [16]. Our previous observations also demonstrated that the severity of NFALD is associated with the presence of high-risk plaques, suggesting a close relationship between NAFLD and coronary atherosclerosis.

Although the association between NAFLD and CAD has previously been mentioned, the relationship between the progression of coronary atherosclerosis and NAFLD remains undetermined. Additionally, several fibrosis score indices have been used to determine the severity of NAFLD, but there is limited information as to which one is a better predictor for atherosclerosis progression. The current study aimed to investigate whether NAFLD indices can help identify patients at risk of CAD progression, especially among those with low CAC scores, and the interactions between other risk factors.

## 2. Methods

### 2.1. Study Population

The CCTA database of the healthcare center in Taipei Veterans General Hospital from 2015 to 2019 was retrospectively reviewed. Among the 1781 asymptomatic subjects who had undergone CCTA for their annual health examination, a total of 243 patients (193 male, 59.8 ± 8.7 years old) had undergone repeated follow-up CCTA. Those with a baseline CAC < 100 and who had undergone repeated CCTA studies were enrolled for analysis. To avoid the interference of pre-existing severe CAD or underlying liver disease, subjects with pre-existing alcoholism, cardiovascular disease (CVD), liver cirrhosis, and severe fatty liver disease were excluded. Eventually, 165 patients (121 male, mean age 58.5 ± 8.6 years) were enrolled in our study. Each subject was given a questionnaire regarding their past medical and surgical history, current medications, and drinking and/or smoking habits. A history of CV disease was defined as previous CAD, myocardial infarction, or cerebrovascular event. DM was defined as fasting glucose ≥ 126 mg/dL and/or glycosylated hemoglobin (HbA1c) level ≥ 6.5% and/or the use of hypoglycemic drugs. Hypertension was defined as a systolic and/or diastolic blood pressure (BP) ≥ 140/90 mmHg and/or the use of antihypertensive drugs. The diagnosis of liver cirrhosis and severe fatty liver disease was established via abdominal sonography. The study was conducted in accordance with the ethical guidelines of the declaration of Helsinki and was approved by the Institutional Review Board of Taipei Veterans General Hospital (IRB number 2018-04-006AC).

### 2.2. Biochemical Markers and Liver Fibrosis Score Measurement

Biochemical parameters were obtained as part of the patient’s health examination survey before the CCTA scans using a TBA-c16000 automatic analyzer (Toshiba Medical Systems, Tochigi, Japan) following overnight fasting. Biochemical parameters, including fasting glucose; lipid profile (triglycerides, high- and low-density lipoprotein cholesterol (HDL and LDL, respectively), and total cholesterol (TC)); kidney function (blood urea nitrogen, creatinine, urate, and estimated glomerular filtration rate (eGFR) levels) and liver function (aspartate transaminase, alanine aminotransferase, alkaline phosphate, and gamma-glutamyl transferase (AST, ALT, ALKP, and GGT, respectively))-related parameters; total and direct bilirubin values; and serum albumin. Noninvasive liver fibrosis scores (LFS), including nonalcoholic fatty liver disease fibrosis score (NFS), fibrosis-4 index (FIB-4), AST to platelet ratio index (APRI), and Forns score, were calculated as follows: (1) NFS = −1.675 + 0.037 × age (years) + 0.094 × body mass index ((BMI); kg/m^2^) + 1.13 × impaired fasting glucose (IFG)/diabetes (yes = 1, no = 0) + 0.99 × AST/ALT ratio −0.013 × platelet count (× 10^9^/L) − 0.66 × albumin (g/dL); (2) FIB-4 score = (age (years) × AST (U/L))/(platelet count (× 10^9^/L) × ALT (U/L)^1/2^): (3) APRI = (AST/upper normal limit of AST)/platelet count (× 10^9^/L) × 100: and (4) Forns score = (7.811 − 3.131 × ln (number of platelets) × 0.781 ln (GGTP [U/L]) + 3.467 × ln (age [years]) − 0.014 (cholesterol [mg/dL])) [17].

### 2.3. CAC Score and Plaque Burden Measurement

CCTA was performed at TVGH with a multiple detector computed tomography scanner (Definition Flash, Siemens Healthineers, Erlangen, Germany). Medications, such as beta-blockers or calcium channel blockers, were given to patients to maintain acceptable heart rates before each CCTA process. We began the scanning sequence at approximately 1 cm above the left main coronary artery. Routine CCTA parameters were set at 120 Kv and 60 Ma and adjusted for patient body size. With a 350 ms gantry rotation time, we were able to achieve a temporal resolution of 230 msec using the half-scan reconstruction method. CCTA was performed using retrospective gated helical scanning with the parameters set at 64 × 0.5–128 × 0.625 mm collimation, 270–350 msec gantry rotation time, and 80–135 Kv according to patient body size. The bolus-tracking method was used for imaging. Each CCTA procedure was performed after injecting 50–100 cc of iodinated contrast medium (Iopamiro370, Bracco Imaging SpA, Milan, Italy; Ultravist 370, Bayer Pharma AG, Berlin, Germany) at a rate of 4.5 to 5.0 cc/s followed by 50 cc of normal saline at a rate of 5.5 cc/s based on the patient’s body size and renal function. A similar protocol was reported in our previous study [18]. The best phase was selected by an automatic system; if the image quality was suboptimal, a certified CCTA technician would manually reconstruct the phase with the best possible image using images with slice thicknesses of 0.75 and 0.9 mm at 0.45 mm intervals. All images were transferred to an external workstation (EBW, Amsterdam, Netherlands) for analysis. Detailed plaque morphology and the degree of coronary artery stenosis were assessed based on previous guidelines. The coronary artery calcium (CAC) score was calculated using the Agatston method. In addition to CAC scores, segment stenosis scores (SSS) were measured to quantify plaque burden and used to represent the overall extent of coronary artery plaques. The anatomical locations of coronary arteries were divided into 16 segments, and each individual coronary segment stenosis was scored from 0 to 3 points based on the extent of coronary artery luminal diameter obstruction (score of 0 signified ≤30% stenosis, score of 1 signified 30–49% stenosis, score of 2 signified 50–69% stenosis, and score of 3 signified ≥70% stenosis). The scores from all 16 individual segments were added together to yield a total score ranging from 0 to 48 [19]. An example of an SSS and CAC calculation is shown in Figure 1.

### 2.4. Statistical Analysis

Because coronary atherosclerosis still progresses despite low CAC scores, SSS changes according to CCTA were used in our current study to represent atherosclerosis involvement and evaluate disease severity more comprehensively. CAD progression was defined as the progression of SSS score. The SSS from the first and second CCTA studies were calculated, and disease progression was defined as any increase in SSS from baseline to follow-up CCTA [20,21]. Multivariable logistic regression analyses using the backward selection method were conducted to evaluate the adjusted odds ratios (ORs) and 95% confidence intervals (CIs) of the association between liver fibrosis markers and the progression of SSS. Variables with a *p*-value < 0.1 in univariable logistic regression were included in the multivariable logistic regression analysis. To validate the backward selection model, we performed the Hosmer and Lemeshow goodness of fit test and determined its area under the curve (AUC). The capability of NFS to predict coronary SSS progression was investigated by using receiver operating characteristic (ROC) curve analyses. The area under the curve and 95% CI were calculated, and the optimal cutoff values were obtained by calculating the Youden’s index. Continuous variables were expressed as mean ± standard deviation. Categorical variables were expressed as frequencies and percentages. All statistical analyses were performed with SPSS software version 25.0 for Windows (IBM, Inc. Chicago, IL, USA). Statistical significance was defined as *p* < 0.05.

## 3. Results

### 3.1. Baseline Characteristics of the Study Population

The baseline characteristics of 165 patients (121 male, mean age 58.5 ± 8.6 years) who had CAC < 100 and had undergone repeated CCTA studies are presented in Table 1. Among these 165 subjects, atherosclerotic plaques were found in 74 (44.8%), while 91 (55.2%) subjects were plaque free at baseline. Those with coronary plaques were older and had a higher percentage of hypertension and/or DM. Baseline lipid profiles were similar between groups with low CAC, but those patients with plaques had higher blood pressure, higher CAC scores, and higher liver fibrosis scores, including Forns score, Fib-4, and NFS.

### 3.2. Factors Associated with Plaque Progression

During the medium follow-up interval of 24.7 months, 60 (39.4%) patients experienced SSS progression, while the remaining 105 (63.6%) patients had no CAD progression. Table 2 shows the baseline difference between those with and without plaque progression. Subjects with coronary atherosclerosis progression were male (predominately) and had higher SBP, DBP, Forns score, NFS, and baseline CAC. According to the multivariate analysis, the male gender (OR 3.055, 95% CI 1.181–7.902); DBP (OR 1.051, 95% CI 1.011–1.093); and NFS liver fibrosis score (OR 1.674, 95% CI 1.169–2.397) were independently associated with coronary SSS progression (Table 3). The Hosmer and Lemeshow test yielded a chi-squared value = 3.686; *p* = 0.884. Using the backward selection method, gender, NFS, and DBP were selected in the multivariate model with a calculated AUC of 0.739 (95% CI 0.660–0.818), as shown in Figure 2.

A subgroup analysis assessing whether the association between NFS, diastolic BP, and SSS progression varied across the different subgroups was performed. The results showed that the association between NFS/DBP and SSS progression were similar in all subgroups, confirming the important role of the NFS fatty liver fibrosis score and diastolic BP in plaque progression during the early stage of atherosclerosis (Figure 3, Appendix A.).

## 4. Discussion

In this retrospective cohort study of 165 patients with a low CAD risk, two important findings were obtained: (1) in addition to traditional risks, such as age, male gender, DM, and hypertension, elevated fatty liver scores, including NFS, Fib-4, and Forns scores, were observed in subjects with mild atherosclerosis and in the initial stages of CAD; and (2) the male gender, DBP, and NFS are independent factors associated with atherosclerotic plaque progression in these low-coronary-calcium-score subjects.

As shown in the PESA (Progression of Early Subclinical Atherosclerosis) study, subclinical atherosclerosis (plaque or coronary artery calcification) was detected in 49.7% of subjects without a cardiovascular risk factor, and a search for a predictor associated with plaque and plaque progression is crucial in CAD prevention [22,23]. It is assumed that the progression of vascular atherosclerosis is associated with multiple traditional risk factors, such as age, gender, hypertension, DM, dyslipidemia, cigarette smoking, and obesity [24]. However, the contributing role of NAFLD, which is frequently observed in high-CV-risk patients, to atherosclerosis remains controversial. NAFLD is the most common chronic liver disease in the developed world [25]. About 20% of patients develop the more serious steatohepatitis (NASH), which is becoming epidemic due to the increase in the rates of obesity and metabolic disease. NASH is a complex disease with multisystem involvement, including diabetes, CV diseases, and metabolic syndrome [26]. Previous studies have shown that NAFLD is associated with the presence, severity, and progression of atherosclerosis [27,28,29,30]. Indeed, the primary cause of death in NAFLD patients is actually due to atherosclerotic CV diseases [31,32]. NASH is a progressive form of NAFLD wherein inflammation causes liver damage and scarring (fibrosis). Recent studies have found that the liver fibrosis score (LFS), which reliably represents the severity of NAFLD liver fibrosis, is also a good marker of atherosclerosis [13,33]. Our previous work demonstrated that NAFLD appears to be associated with plaque burden and the presence of vulnerable plaque, suggesting an association between liver steatosis and coronary atherosclerosis [18]. Our study clearly showed a correlation between the presence of plaques and more severe liver steatosis. Higashiura et al. found that the LFS was an independent predictor for the development of ischemic heart disease among 13,448 subjects in a 10-year follow-up study [34]. Furthermore, Schonmann et al. demonstrated that the LFS could be an independent predictor of cardiovascular morbidity and mortality in the general population, even in those without NAFLD [13]. These results suggest that the LFS is a useful marker of cardiometabolic risk evaluation, even for patients without an NAFLD diagnosis.

In this study, we found that those who had coronary plaques were older; predominantly male; and had a higher percentage of smoking habits, diabetes, and hypertension. Notably, subjects with basal coronary plaques had higher fatty fibrosis scores (NFS, Fib-4, and Forns score), suggesting that liver fibrosis scores could be used for the early identification of coronary plaques and those who may be at risk of developing adverse events in the future. In the later follow-up, the NFS liver fibrosis score, male gender, and DBP in particular were found to be independently associated with coronary atherosclerosis progression, and the association between the NFS and plaque progression remained significant across various subgroups. The NFS, which is calculated from metabolic biomarkers, is based on the idea that DM status may be a more suitable tool to assess the degree of metabolic dysfunction. Several previous studies have also shown the superior prognostic value of NFS over other liver fibrosis markers [35,36,37].

To the best of our knowledge, this study is the first investigation to explore the association between CAD progression and various biomarkers in the low-risk population. Our study adds to the growing amount of evidence that the LFS is an important risk indicator to further risk-stratify patients with a low CAD risk, especially in the general population. Lee et al. reported that the LFS was associated with the progression of the CAC score in 293 NAFLD patients [38]. Our findings extend this association to the general population. The close link between LFS and CAD depends on many potential mechanisms. First, NAFLD shares many risk factors with CAD. Indeed, the Framingham risk score and NAFLD are highly correlated [39]. Second, LFS and CAD share many pathophysiologic mechanisms, including systemic inflammation, endothelial dysfunction, hepatic insulin resistance, oxidative stress, and altered lipid metabolism [40]. With the previous evidence as a basis, our findings suggest that the cardiometabolic disease burden, as identified by the NFS, may be key to distinguishing patients who are at risk of progression versus those who are not. This evidence suggests that LFS is an easy and cheap risk indicator for gauging the metabolic disease burden of patients with a low CAD risk.

In addition to the NFS, it was demonstrated that blood pressure, especially DBP, is associated with the presence and progression of CAD. An association between subclinical CAD and baseline blood pressure has been reported in the general population without hypertension [41,42]. For patients with CAD, baseline blood pressure is closely associated with severity and outcome [43,44]. Sipahi et al. showed that baseline SBP was independently associated with the progression of coronary atheroma burden in 274 patients who completed repeat intravascular ultrasound studies, while our study showed that DBP is an independent predictor of CAD progression [45]. This finding may have resulted from the level of SBP in our low-risk population, which was well within the “sweet spot” of 120 to 140 mm Hg and corresponded to no net progression or regression of coronary disease, as shown in a study by Sipahi et al. A larger study by Vidal-Petiot et al. reported that DBP is independently associated with a poor CV outcome in all stratified SBP levels and vice versa [46]. These findings indicate the importance of blood pressure control for CAD risk reduction. Indeed, a previous interventional trial showed that the relationship between blood pressure and CAD progression was independent of the study treatments involving angiotensin-converting enzyme inhibitors or calcium-channel blockers [47]. This finding demonstrates that the absolute blood pressure attained is more crucial than the selected antihypertensive drug [45].

## 5. Study Limitations

This study had several potential limitations. First, the study was retrospective; therefore, causality cannot be determined. Second, our study had a small sample population; thus, some biomarkers with less significant effects may not have been detected. Third, detailed medication regimens and information about patient adherence could not be obtained. Thus, medication effects could not be assessed in our study. Fourth, information regarding inflammatory cytokines, such as tumor necrosis factor-alpha (TNFα), interleukin-1beta (IL-1β), and IL-6, which are also important for CAD and NAFLD, was unavailable for our study. However, the LFSs were calculated from routinely measured lab data, making them more practical for clinical use.

## 6. Conclusions

A higher baseline blood pressure and increased liver fibrosis markers were associated with the presence of CAD plaques in subjects with mild atherosclerosis and early-stage CAD. For disease progression, the male gender, DBP, and NFS were independently associated with atherosclerosis plaque progression in subjects with subclinical atherosclerosis. The association between the NFS NAFLD marker and CAD progression is a novel finding that adds to the previously known traditional risk factors. Our results indicate that cardiometabolic disease burden, as identified by NFS, may be key to further stratifying low-risk patients who are at risk of CAD progression.

## Figures and Tables

**Figure 1 nutrients-14-03163-f001:**
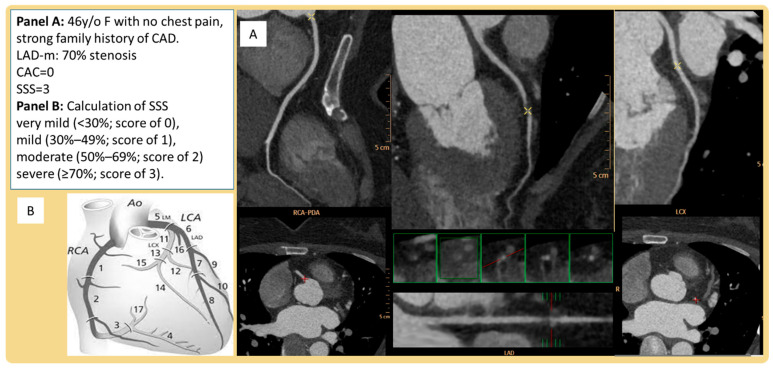
Example of SSS calculation. CAD = coronary artery disease, LAD = left anterior descending artery, SSS = segment stenosis scores.

**Figure 2 nutrients-14-03163-f002:**
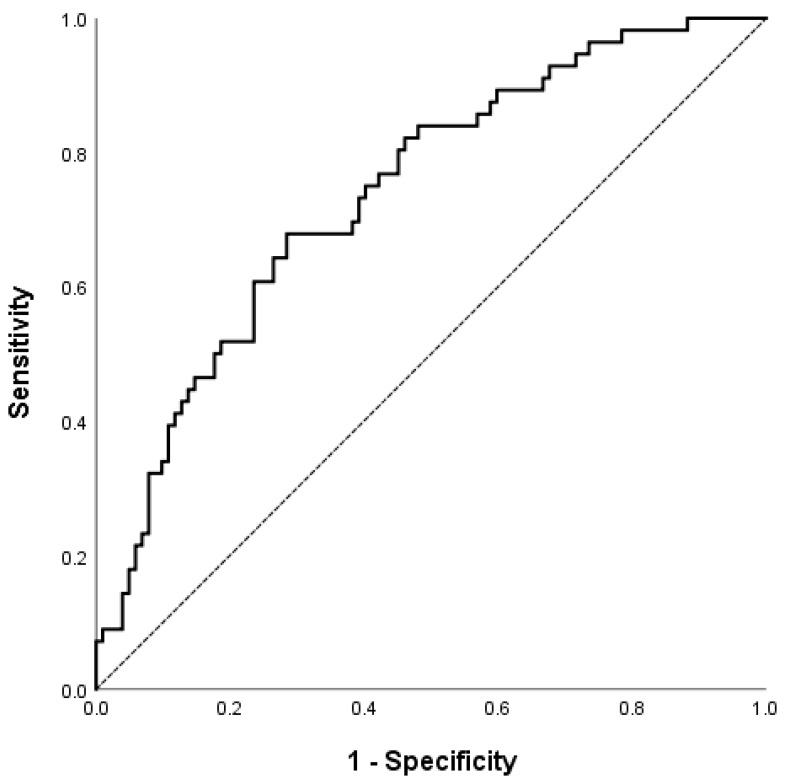
ROC curve of the multivariate logistic regression model for CAD progression; AUC was 0.739 (95% CI 0.660–0.818), *p* < 0.001. The dash line represent the line of no-discrimination.

**Figure 3 nutrients-14-03163-f003:**
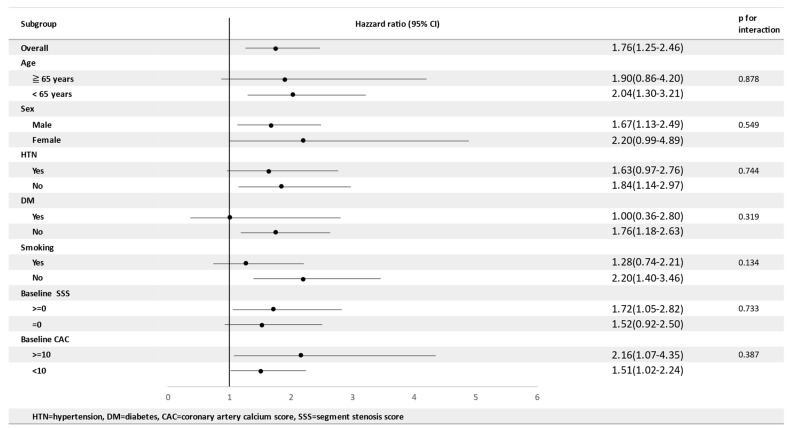
Interaction between NFS and the progression of atherosclerotic plaques in different subgroups.

**Table 1 nutrients-14-03163-t001:** Baseline characteristics of low-risk patients with and without initial SSS (*n* = 165).

Characteristics	Overall *n* = 165	SSS Zero *n* = 74	SSS Positive *n* = 91	*p*-Value
Male Gender (*n*, %)	121 (73.3%)	59 (64.8%)	62 (83.8%)	0.008
Age (y/o)	58.5 ± 8.6	56.7 ± 8.8	60.8 ± 7.9	0.002
BMI	25.1 ± 3.1	24.8 ± 3.2	25.4 ± 2.9	0.197
Hypertension (*n*, %)	55 (34.8%)	21 (23.9%)	34 (48.6%)	0.001
Diabetes (*n*, %)	16 (10.1%)	4 (4.5%)	12 (17.1%)	0.015
Dyslipidemia (*n*, %)	46 (29.1%)	22 (25.0%)	24 (34.3%)	0.221
Smoking (*n*, %)	51 (31.1%)	29 (31.9%)	22 (30.1%)	0.866
Drinking (*n*, %)	71 (43.6%)	40 (44.0%)	31 (43.1%)	1.000
Cholesterol	206.6 ± 38.3	209.8 ± 33.9	202.6 ± 42.9	0.226
Triglyceride	143.3 ± 80.5	143.9 ± 85.3	142.6 ± 74.9	0.914
Uric acid	6.5 ± 1.5	6.3 ± 1.4	6.8 ± 1.5	0.057
HDL	46.6 ± 13.4	48.0 ± 14.7	44.8 ± 11.5	0.128
LDL	129.9 ± 33.3	131.1 ± 31.3	128.4 ± 35.7	0.608
Glucose	98.8 ± 24.4	96.2 ± 20.7	101.9 ± 28.2	0.142
AST	25.9 ± 0.0	25.5 ± 0.0	26.4 ± 0.0	0.539
ALT	30.6 ± 0.0	31.3 ± 0.0	29.7 ± 0.0	0.569
ALK-P	64.0 ± 17.8	64.7 ± 16.6	63.2 ± 19.2	0.581
Total bilirubin	1.1 ± 0.5	1.1 ± 0.4	1.1 ± 0.5	0.931
Direct bilirubin	0.3 ± 0.1	0.3 ± 0.1	0.3 ± 0.1	0.857
GGT	33.0 ± 25.7	35.7 ± 29.2	29.5 ± 20.3	0.112
LDH	176.6 ± 51.1	173.2 ± 29.2	180.7 ± 69.1	0.350
Albumin	4.5 ± 0.2	4.5 ± 0.2	4.4 ± 0.3	0.342
Total protein	7.3 ± 0.5	7.3 ± 0.5	7.3 ± 0.5	0.673
BUN	12.9 ± 3.8	12.6 ± 3.6	13.3 ± 4.1	0.262
Creatinine	0.9 ± 0.2	0.9 ± 0.2	0.9 ± 0.2	0.111
eGFR	82.5 ± 12.1	82.9 ± 11.1	82.1 ± 13.3	0.675
HsCRP	0.2 ± 0.3	0.3 ± 0.4	0.2 ± 0.1	0.589
CK	95.7 ± 0.0	79.6 ± 0.0	115.3 ± 0.0	0.235
Na	141.3 ± 1.9	141.4 ± 2.0	141.1 ± 1.8	0.294
K	4.1 ± 0.3	4.1 ± 0.4	4.1 ± 0.3	0.389
SBP	122.8 ± 17.6	118.5 ± 17.7	128.0 ± 16.2	0.001
DBP	77.5 ± 10.4	75.6 ± 10.4	79.8 ± 9.9	0.010
APRI	0.3 ± 0.1	0.3 ± 0.1	0.3 ± 0.2	0.273
Forns score	4.9 ± 0.7	4.8 ± 0.6	5.0 ± 0.8	0.035
FIB-4	1.3 ± 0.6	1.3 ± 0.5	1.4 ± 0.6	0.042
NFS	−2.0 ± 1.1	−2.2 ± 1.0	−1.7 ± 1.1	0.003
LVEF	62.3 ± 7.0	61.1 ± 6.7	63.7 ± 7.0	0.033
Baseline CAC	12.9 ± 23.6	1.0 ± 5.0	27.5 ± 28.7	<0.001
Baseline SSS	1.2 ± 2.0	0.0 ± 0.0	2.7 ± 2.2	<0.001

BMI = body mass index, HDL = high-density lipoprotein, LDL = low-density lipoprotein, AST = aspartate transaminase, ALT = alanine aminotransferase, ALK-P = alkaline phosphatase, GGT= gamma-glutamyl transferase, LDH = lactate dehydrogenase, BUN = blood urea nitrogen, eGFR = estimated glomerular filtration rate, HsCRP = high-sensitivity C-reactive protein, CK = creatine kinase, SBP = systolic blood pressure, DBP = diastolic blood pressure, NFS = nonalcoholic fatty liver disease fibrosis score, FIB-4 = fibrosis-4 index, APRI = AST to platelet ratio index, LVEF = left ventricular ejection fraction, CAC = coronary artery calcium, SSS = segment stenosis score.

**Table 2 nutrients-14-03163-t002:** Baseline characteristics of low-risk patients with and without SSS progression (*n* = 165).

Characteristics	Overall *n* = 165	No SSS Progression *n* = 105	SSS Progression *n* = 60	*p*-Value
Male Gender (*n*, %)	121 (73.3%)	69 (65.7%)	52 (86.7%)	0.003
Age (y/o)	58.5 ± 0.7	57.3 ± 8.8	60.6 ± 8.0	0.021
BMI	25.1 ± 0.2	24.9 ± 3.1	25.3 ± 3.1	0.489
Hypertension (*n*, %)	55 (34.8%)	32 (31.4%)	23 (41.1%)	0.228
Diabetes (*n*, %)	16 (10.1%)	6 (5.9%)	10 (17.9%)	0.026
Dyslipidemia (*n*, %)	46 (29.1%)	30 (29.4%)	16 (28.6%)	1.000
Smoking (*n*, %)	51 (31.1%)	29 (27.9%)	22 (36.7%)	0.294
Drinking (*n*, %)	71 (43.6%)	43 (41.7%)	28 (46.7%)	0.624
Cholesterol	206.6 ± 3.0	208.7 ± 35.3	202.9 ± 43.0	0.352
Triglyceride	143.3 ± 6.3	144.3 ± 84.5	141.6 ± 73.8	0.832
Uric acid	6.5 ± 0.1	6.4 ± 1.5	6.7 ± 1.6	0.282
HDL	46.6 ± 1.0	47.9 ± 14.2	44.3 ± 11.8	0.096
LDL	129.9 ± 2.6	130.2 ± 31.3	129.4 ± 36.8	0.881
Glucose	98.8 ± 1.9	96.7 ± 20.1	102.3 ± 30.4	0.205
AST	25.9 ± 0.8	25.9 ± 0.0	26.0 ± 0.0	0.931
ALT	30.6 ± 1.3	30.5 ± 0.0	30.8 ± 0.0	0.930
ALK-P	64.0 ± 1.4	63.1 ± 15.2	65.5 ± 21.6	0.407
Total bilirubin	1.1 ± 0.0	1.0 ± 0.4	1.1 ± 0.6	0.260
Direct bilirubin	0.3 ± 0.0	0.3 ± 0.1	0.4 ± 0.1	0.420
GGT	33.0 ± 2.0	32.5 ± 25.4	33.8 ± 26.6	0.755
LDH	176.6 ± 4.0	171.9 ± 27.2	185.0 ± 76.8	0.114
Albumin	4.5 ± 0.0	4.5 ± 0.2	4.4 ± 0.2	0.246
Total protein	7.3 ± 0.0	7.3 ± 0.4	7.3 ± 0.5	0.730
BUN	12.9 ± 0.3	12.6 ± 3.8	13.5 ± 3.8	0.184
Creatinine	0.9 ± 0.0	0.9 ± 0.2	0.9 ± 0.1	0.440
eGFR	82.5 ± 0.9	82.0 ± 11.7	83.5 ± 12.7	0.431
HsCRP	0.2 ± 0.1	0.2 ± 0.4	0.2 ± 0.1	0.813
CK	95.7 ± 14.9	79.8 ± 0.0	124.0 ± 0.0	0.280
Na	141.3 ± 0.2	141.3 ± 1.9	141.1 ± 1.9	0.551
K	4.1 ± 0.0	4.1 ± 0.4	4.1 ± 0.3	0.970
SBP	122.8 ± 1.4	119.4 ± 16.1	128.7 ± 18.7	0.001
DBP	77.5 ± 0.8	75.4 ± 10.1	81.0 ± 9.9	0.001
APRI	0.3 ± 0.0	0.3 ± 0.1	0.3 ± 0.2	0.182
Forns score	4.9 ± 0.1	4.8 ± 0.6	5.1 ± 0.8	0.006
FIB-4	1.3 ± 0.0	1.3 ± 0.5	1.5 ± 0.6	0.056
NFS	−2.0 ± 0.1	−2.2 ± 1.0	−1.6 ± 1.0	0.001
LVEF	62.3 ± 0.6	62.1 ± 6.9	62.7 ± 7.1	0.669
Baseline CAC	12.9 ± 1.8	8.6 ± 19.2	20.4 ± 28.4	0.002
Baseline SSS	1.2 ± 0.2	1.0 ± 1.9	1.5 ± 2.0	0.059

BMI = body mass index, HDL = high-density lipoprotein, LDL = low-density lipoprotein, AST = aspartate transaminase, ALT = alanine aminotransferase, ALK-P = alkaline phosphatase, GGT= gamma-glutamyl transferase, LDH = lactate dehydrogenase, BUN = blood urea nitrogen, eGFR = estimated glomerular filtration rate, HsCRP = high-sensitivity C-reactive protein, CK = creatine kinase, SBP = systolic blood pressure, DBP = diastolic blood pressure, NFS = nonalcoholic fatty liver disease fibrosis score, FIB-4 = fibrosis-4 index, APRI = AST to platelet ratio index, LVEF = left ventricular ejection fraction, CAC = coronary artery calcium, SSS = segment stenosis score.

**Table 3 nutrients-14-03163-t003:** Association between baseline biomarkers and SSS progression.

	Unadjusted (Model 1)		Adjusted * (Model 2)	
	HR (95% CI)	*p*-Value	HR (95% CI)	*p*-Value
Age	1.05 (1.01–1.09)	0.023		
Gender	3.39 (1.45–7.91)	0.005	3.055 (1.181–7.902)	0.021
Hypertension	1.52 (0.77–3.00)	0.222		
Diabetes	3.48 (1.19–10.15)	0.023		
Hyperlipidemia	0.96 (0.47–1.97)	0.911		
BMI	1.04 (0.94–1.15)	0.486		
SBP	1.03 (1.01–1.05)	0.001		
DBP	1.06 (1.02–1.09)	0.001	1.051 (1.011–1.093)	0.013
Smoking	1.50 (0.76–2.95)	0.243		
Drinking	1.22 (0.64–2.32)	0.541		
Cholesterol	1.00 (0.99–1.00)	0.350		
Triglyceride	1.00 (1.00–1.00)	0.830		
Uric acid	1.12 (0.91–1.39)	0.281		
HDL	0.98 (0.95–1.00)	0.099		
LDL	1.00 (0.99–1.01)	0.880		
GLU	1.01 (1.00–1.02)	0.170		
AST	1.00 (0.97–1.03)	0.930		
ALT	1.00 (0.98–1.02)	0.930		
ALK-P	1.01 (0.99–1.03)	0.406		
Total bilirubin	1.47 (0.75–2.88)	0.261		
Direct bilirubin	2.92 (0.22–38.99)	0.418		
GGT	1.00 (0.99–1.01)	0.753		
LDH	1.01 (1.00–1.01)	0.184		
Albumin	0.46 (0.12–1.73)	0.250		
Total protein	0.88 (0.44–1.77)	0.728		
BUN	1.06 (0.97–1.15)	0.185		
Creatinine	2.04 (0.31–13.56)	0.460		
HsCRP	0.62 (0.01–27.93)	0.803		
CK	1.00 (1.00–1.01)	0.379		
NA	0.95 (0.81–1.12)	0.549		
K	1.02 (0.40–2.57)	0.970		
LVEF	1.01 (0.96–1.06)	0.666		
NFS	1.76 (1.25–2.46)	0.001	1.674 (1.169–2.397)	0.005
FIB-4	1.70 (0.98–2.95)	0.059		
APRI	4.54 (0.47–43.40)	0.189		
Forns score	1.91 (1.19–3.06)	0.007		
Baseline CAC	1.02 (1.01–1.04)	0.003		
Baseline SSS	1.13 (0.96–1.32)	0.147		

BMI = body mass index, HDL = high-density lipoprotein, LDL = low-density lipoprotein, AST = aspartate transaminase, ALT = alanine aminotransferase, ALK-P = alkaline phosphatase, GGT= gamma-glutamyl transferase, LDH = lactate dehydrogenase, BUN = blood urea nitrogen, eGFR = estimated glomerular filtration rate, HsCRP = high-sensitivity C-reactive protein, CK = creatine kinase, SBP = systolic blood pressure, DBP = diastolic blood pressure, NFS = nonalcoholic fatty liver disease fibrosis score, FIB-4 = fibrosis-4 index, APRI = AST to platelet ratio index, LVEF = left ventricular ejection fraction, CAC = coronary artery calcium, SSS = segment stenosis score. * Adjusted model: covariate selection for the multivariate analysis was based on *p* < 0.05 in univariate analysis, with a logistic regression model and a backward elimination procedure.

## Data Availability

The data of this study are available upon request to the corresponding author.

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
