# Peer review of "Association between Coronary Artery Plaque Progression and Liver Fibrosis Biomarkers in Population with Low Calcium Scores"

_nutrients, 2022, doi:10.3390/nu14153163_

Round 1
Reviewer 1 Report
One of the concerns is that authors did not determine the levels of inflammatory cytokines, such as tumor necrosis factor-alpha (TNFα), interleukin-1beta (IL-1β) and IL-6 and fat derived adipokine. The levels of TNFα, IL-1β and IL-6 would be indicative of nonalcoholic steatohepatitis (NASH). NAFLD progress to NASH and then leads to cell death in liver causing cirrhosis and fibrosis. Is there any relationship of NASH with the severity of CAD? This aspect has been ignored.
Line 110: The IRB number is not mentioned “Institutional Review Board of Taipei Veterans General Hospital. (IRB number _____)”
Author Response
1. One of the concerns is that authors did not determine the levels of inflammatory cytokines, such as tumor necrosis factor-alpha (TNFα), interleukin-1beta (IL-1β) and IL-6 and fat derived adipokine. The levels of TNFα, IL-1β and IL-6 would be indicative of nonalcoholic steatohepatitis (NASH). NAFLD progress to NASH and then leads to cell death in liver causing cirrhosis and fibrosis. Is there any relationship of NASH with the severity of CAD? This aspect has been ignored.
Reply: thank you for this fine comment. Regrettably, tumor necrosis factor-alpha (TNFα), interleukin-1beta (IL-1β), and IL-6 and fat-derived adipokine data were unavailable. Thus we couldn't provide the relationship between inflammatory cytokine and CAD progression in our study. We have added the lack of inflammatory cytokine information to the limitations of our study. Line 362-366
Line 110: The IRB number is not mentioned “Institutional Review Board of Taipei Veterans General Hospital. (IRB number _____)”
Reply: Thank you for pointing this out. We apologize for not presenting the IRB number in the first place. The IRB number was 2018-04-006AC for out study. Line 108-111.
Reviewer 2 Report
The novelty of the study should be more clearly emphasized at the end of the introduction.
More precise data on all patients included into the study (age, gender etc.) should be added in the chapter 2.1.
Whether the research required approval from a bioethical committee. If not - it should be clearly underlined, if so - give the consent number and the name of the committee.
If the table (Figure 3) is presented in the manuscript, it is not necessary to add the same table in supplementary files
In conclusion eventually medical usage of study should be mentioned, and novelty of observation should be underlined more clearly
Author Response
Response to reviewer 2:
The novelty of the study should be more clearly emphasized at the end of the introduction.
Reply: Thank you for your comment, we have modified our introduction in the revised manuscript. Line 80-87.
More precise data on all patients included into the study (age, gender etc.) should be added in the chapter 2.1.
Reply: we have added a more detailed data of patient enrollment in the revised manuscript. Line93, 98-99.
Whether the research required approval from a bioethical committee. If not - it should be clearly underlined, if so - give the consent number and the name of the committee.
Reply: Thank you for your comment, we apologize for not providing the IRB information before submitting this article. We have added the IRB information in the revised manuscript. Institutional Review Board of Taipei Veterans General Hospital. (IRB number 2018-04-006AC). Line108-111
If the table (Figure 3) is presented in the manuscript, it is not necessary to add the same table in supplementary files
Reply: Thank you for your comment. Figure 3 represent the interaction between NFS and the progression of atherosclerotic plaques in different subgroups while supplemental figure 1 represent the interaction between DBP and the progression of atherosclerotic plaques in different subgroups. However, the editors seemed to have removed all figure legends. For clarity, we have removed supplemental Figure 1.
In conclusion eventually medical usage of study should be mentioned, and novelty of observation should be underlined more clearly
Reply: Thank you for your comment, we have modified our conclusion in the revised manuscript. Line 368-372.